Rapid and convenient detection of SARS-CoV-2 using a colorimetric triple-target reverse transcription loop-mediated isothermal amplification method

Yang Zhu zhuyang@wnmc.edu.cn 1
Liu Nicole Y. 1
Zhu Zhiwei 2
Xiao Minmin 3
Zhong Shuzhi 4
Xue Qiqi 2
Nie Lina 3
Zhao Jinhong zhaojh@wnmc.edu.cn 30363754@qq.com 2
1 Department of Medical Microbiology and Immunology, Wannan Medical College , Wuhu , Anhui , China
2 Department of Parasitology, Wannan Medical College , Wuhu , Anhui , China
3 Clinical Laboratory, The Second People’s Hospital of Wuhu City , Wuhu , Anhui , China
4 Department of Histology and Embryology, Wannan Medical College , Wuhu , Anhui , China
Gould Gwyn
Electronic publication date: 2022 Oct 10
Publication date: 2022
Volume: 10
Electronic Location ID: e14121
Received 2022 Jun 29; Accepted 2022 Sep 5
Copyright: ©2022 Yang et al.
Copyright year: 2022
Copyright holder: Yang et al.
License: This is an open access article distributed under the terms of the Creative Commons Attribution License, which permits unrestricted use, distribution, reproduction and adaptation in any medium and for any purpose provided that it is properly attributed. For attribution, the original author(s), title, publication source (PeerJ) and either DOI or URL of the article must be cited.
License URL: https://creativecommons.org/licenses/by/4.0/

Keywords: SARS-CoV-2, RT-LAMP, Triple target, Colorimetric

Funding: Wuhu Science and Technology Plan Project 2020ms3-9 2021jc2-6 2021yf39 Scientific Research Project of Wannan Medical College wyqnyx202004 WK2021Z07 JXYY202102 Academic Aid Program for Top-Notch Talents in Provincial Universities gxbjZD2020071 Anhui University Natural Science Research Project KJ2020ZD56 This research has been supported by grants from Wuhu Science and Technology Plan Project (No. 2020ms3-9, 2021jc2-6 and 2021yf39), Scientific Research Project of Wannan Medical College (No. wyqnyx202004, WK2021Z07 and JXYY202102), Academic Aid Program for Top-Notch Talents in Provincial Universities (No. gxbjZD2020071), Anhui University Natural Science Research Project (No. KJ2020ZD56). There was no additional external funding received for this study. The funders had no role in study design, data collection and analysis, decision to publish, or preparation of the manuscript.

==============================
Coronavirus Disease 2019 (COVID-19) caused by SARS-CoV-2 poses a significant threat to global public health. Early detection with reliable, fast, and simple assays is crucial to contain the spread of SARS-CoV-2. The real-time reverse transcription-polymerase chain reaction (RT-PCR) assay is currently the gold standard for SARS-CoV-2 detection; however, the reverse transcription loop-mediated isothermal amplification method (RT-LAMP) assay may allow for faster, simpler and cheaper screening of SARS-CoV-2. In this study, the triple-target RT-LAMP assay was first established to simultaneously detect three different target regions (ORF1ab, N and E genes) of SARS-CoV-2. The results revealed that the developed triplex RT-LAMP assay was able to detect down to 11 copies of SARS-CoV-2 RNA per 25 µL reaction, with greater sensitivity than singleplex or duplex RT-LAMP assays. Moreover, two different indicators, hydroxy naphthol blue (HNB) and cresol red, were studied in the colorimetric RT-LAMP assay; our results suggest that both indicators are suitable for RT-LAMP reactions with an obvious color change. In conclusion, our developed triplex colorimetric RT-LAMP assay may be useful for the screening of COVID-19 cases in limited-resource areas.

Introduction

SARS-CoV-2 is a highly pathogenic coronavirus causing COVID-19, which was first reported in December 2019 in Wuhan, China (Wu et al., 2020; Zhou et al., 2020; Zhu et al., 2020). SARS-CoV-2 has spread throughout the world and has resulted in a new global pandemic, as of May 2022, over 529 million infections and more than 6 million deaths reported (WHO, 2021). As cases are usually identified through large-scale local screening of individuals during regional COVID-19 outbreaks, it is important to develop more rapid and convenient assays for SARS-CoV-2 detection (Li et al., 2021; Yoon et al., 2022). Real-time RT-PCR with high sensitivity and specificity is the most common assay currently used to detect SARS-CoV-2, and many approved commercial real-time RT-PCR kits are available and widely used in public health and clinical laboratories. However, real-time RT-PCR has some limitations, such as the need for expensive real-time PCR instruments and well-trained personnel, as well as a long ‘samples to results’ time (usually 4 h). These issues hamper the use of PCR-based methods (Lu et al., 2020).

As one of the novel nucleic acid (DNA or RNA) isothermal amplification methods, LAMP assays have several advantages and are suitable for point of care testing (POCT) and field applications (Notomi et al., 2015). First, LAMP can be carried out by individuals without special training or expensive equipment, which makes it applicable for SARS-CoV-2 screening in resource-limited regions. Moreover, samples without special nucleic acid isolation can be directly used as the templates in LAMP reactions, and the estimated cost for each RT-LAMP reaction is below two dollars (Rabe & Cepko, 2020; Schermer et al., 2020). Furthermore, the results of the LAMP reaction can be observed by easy-to-see color changes (Goto et al., 2009; Rabe & Cepko, 2020; Tanner, Zhang & Evans Jr, 2015). Last but not least, LAMP assays have been successfully used for detecting emerging pathogens, such as parasites (Ortleppascaris sinensis) (Zhao et al., 2019); bacteria (TB, and Salmonella) (WHO, 2016; Kim et al., 2022); and viruses (HIV, MERS-CoV, and SARS-CoV) (Kim et al., 2019; Lee et al., 2017; Li et al., 2019). Therefore, RT-LAMP assays are of great value for the screening of SARS-CoV-2 in places such as outpatient clinics and in the field, especially in resource-limited regions.

In the study reported here, we successfully developed a triple-target colorimetric RT-LAMP assay for rapid and convenient detection of SARS-CoV-2. First, six sets of LAMP primers were studied based on their amplification performance in real-time RT-LAMP reaction using SARS-CoV-2 genome RNA standard as the templates, different combination of LAMP primer sets were performed and triplex RT-LAMP assays targeting the ORF1ab, E and N genes of SARS-CoV-2 were first established, which may prevent failure to detect the target due to genetic mutations and improve the accuracy of detection. Second, the developed triplex RT-LAMP assay showed a higher sensitivity than the singleplex or duplex RT-LAMP assay, detecting down to 11 copies of SARS-CoV-2 RNA per reaction. Third, two different indicators, hydroxy naphthol blue (HNB) and cresol red were studied in the RT-LAMP assay; and our results suggested that both indicators were suitable for colorimetric RT-LAMP reactions. In conclusion, the developed triplex colorimetric RT-LAMP assay offers a new promising tool for rapid and convenient screening of SARS-CoV-2 in resource-limited areas around the world.

Materials & Methods

LAMP primers

In this study, the LAMP primer sets targeting the ORF1ab, N, and E genes were constructed based on RT-LAMP assays previously reported by the different laboratories, respectively (Broughton et al., 2020; Dong et al., 2021; Jamwal et al., 2021; Jiang et al., 2020; Nawattanapaiboon et al., 2021; Park et al., 2020; Yu et al., 2021; Zhang & Tanner, 2021). A set of LAMP primers consisting of six primers (F3, B3, FIP, BIP, LF, LB) targeted eight distinct regions of the templates, and all primers were ordered from Sangon Biotech (Shanghai), LAMP primer mixtures (F3/B3 2 µM each; FIP/BIP 16 µM each; LF/LB 4 µM each) were prepared and used for the further RT-LAMP reactions.

Preparation of the different dilutions of SARS-CoV-2 genome RNA standards

Certified reference material of SARS-CoV-2 genome RNA was purchased from the National Reference Material Resource Sharing Platform (http://www.ncrm.org.cn, GBW(E)091099) and the copies number of ORF1ab, E and N genes per reaction were calculated with the instructions. ORF1ab was the largest and the most conserved gene regions within SARS-CoV-2 genome, so we chose ORF1ab as the standard in our studies (Hu et al., 2021). Different gradient dilutions were prepared with EASY Dilution (9160; Takara, Dalian, China), and a panel of SARS-CoV-2 RNA standards ranging from 448 to 4 copies (ORF1ab gene) per reaction was used for further studies (Table 1).

Real-time RT-LAMP reaction

Real-time RT-LAMP assays were performed on the LightCycler 96 real-time PCR system (Roche Diagnostics, Germany) with a WarmStart® LAMP kit (NEB, E1700S) according to the manufacturer’s protocol. The real-time RT-LAMP reaction (25 µl) contained 5 µl SARS-CoV-2 RNA template (448 copies), 12.5 µl 2 × LAMP reaction buffer, 2.5 µl LAMP primer mix, 0.5 µl Dye, and 4.5 µl DEPC-H2O. DEPC-H2O was used as a negative control. The reaction was carried out at 65 °C for 45 min and the fluorescence signals were collected at 30 secs intervals on the SYBR Green channel, followed by melting curve analysis.

Multiplexed real-time RT-LAMP assay

Multiplex real-time RT-LAMP reactions were performed as follows. For dual-target real-time RT-LAMP reaction, the additional LAMP primer mixture replaced 2.5 µl of DEPC-H2O. For triple-target real-time RT-LAMP reaction, the two additional LAMP primer mixtures (2.5 µl each) replaced 4.5 µl of DEPC-H2O. 5µl SARS-CoV-2 RNA template (448 copies) was used for each reaction. DEPC-H2O was used as a negative control.

Colorimetric RT-LAMP reaction

To prepare further usage of the RT-LAMP assay for POCT or field applications, we developed visual detection of the RT-LAMP reaction with colorimetric methods. Hydroxy naphthol blue (H811452-5g; HNB, Macklin), which is a metal-ion sensitive indicator for monitoring the change of Mg2+ ion concentration in LAMP reactions, was added to the RT-LAMP reaction system (E1700S; NEB, Ipswich, MA, USA) as follows: 5 µl SARS-CoV-2 RNA template, 12.5 µl 2 × LAMP reaction buffer, 2.5 µl LAMP primer mix, 1 µl HNB (3mM), and 4 µl DEPC-H2O. WarmStart Colorimetric LAMP 2 × Master Mix (NEB, M1800L) containing cresol red, which is a pH sensitive indicator for determining the drop in pH caused by LAMP amplification, was added to the RT-LAMP reaction as follows: 5 µl SARS-CoV-2 RNA template, 12.5 µl 2 × LAMP reaction buffer, 2.5 µl LAMP primer mix and 5 µl DEPC-H2O. After incubation at 65 °C for 35 min, the positive reaction with HNB led to a color change from violet to blue, and the positive reaction with cresol red led to color change from pink to yellow (Goto et al., 2009; Tanner, Zhang & Evans Jr, 2015).

Table 1 Gradient dilutions and different copies number of SARS-CoV-2 genome RNA standards per reaction.

Dilution	E0	E-1	E-2	1/2E-2	1/4E-2	E-3	E-4	
ORF1ab gene
(Copies/Reaction)	4480	448	44	22	11	4	<1	
N gene
(Copies/Reaction)	8650	865	86	43	21	8	<1	
E gene
(Copies/Reaction)	5300	530	53	26	13	2	<1	

Sensitivity of the triple target RT-LAMP assay for SARS-CoV-2

A panel of SARS-CoV-2 genome RNA standards ranging from 448 to 4 copies (Table 1) was used as the templates to determine the sensitivity of the developed triplex real-time and colorimetric RT-LAMP assay. Each template was performed in triplicate, and the template with the lowest copy number detected positively was defined as the sensitivity of the assay (Dong et al., 2021).

Specificity of the triple target RT-LAMP assay for SARS-CoV-2

First, in silico analyses of the selected LAMP primers were performed to validate the specificity. Moreover, the specificity of the developed RT-LAMP assay was evaluated with RNA isolated from positive clinical samples with some common respiratory viruses (including human seasonal coronavirus (HCoV) types HKU1, NL63, OC43, and 229E; human seasonal influenza A virus subtypes H1N1, and H3N2, influenza B virus; human parainfluenza virus (HPIV) types 1, 2, and 3; human respiratory syncytial virus (RSV) subgroups A and B.

Results

LAMP primers design and selection

Six sets of LAMP primers were designed to detect the SARS-CoV-2 ORF1ab gene (set1 and set2), N gene (set3 and set4), and E gene (set5 and set6), respectively (Fig. 1 and Table 2). After LAMP primers generated, we blasted them in the NCBI (http://blast.ncbi.nlm.nih.gov/Blast.cgi) database to examine and validate their specificity. The analyses showed that these primers were 100% matching with the SARS-CoV-2 genome sequences (Supplemental 1).

Figure 1 Locations of the different LAMP primer sets within the SARS-CoV-2 genome.

Using prepared SARS-CoV-2 genome RNA standard as the template (448 copies), we evaluated the amplification performance of the six sets of LAMP primers in the real-time RT-LAMP assays. All six primer sets generated amplification curves and reached the plateau phase within 40 min, with set3 showing the fastest amplification among all six designed LAMP primer sets (Fig. 2A). Furthermore, set1 (S1) was faster than set2 (S2), both targeting the ORF1ab gene; similarly, set3 (S3) was faster than set4 (S4), both targeting the N gene; and set5 (S5) was faster than set6 (S6), both targeting the E gene (Fig. 2B). Faster amplification is often associated with higher detection sensitivity, so these three LAMP primer sets (S1, S3 and S5) were selected for the further LAMP primer sets combination studies (Dong et al., 2021).

To optimize the RT-LAMP reaction, different reaction times (ranging from 30 to 60 min) were executed in the real-time RT-LAMP assays, using 4 copies of SARS-CoV-2 genome RNA as the template, positive signal shown after 30-min reaction time, and NTC shown positively in 45-min reaction time, so 35-min was selected as the optimal reaction time in order to reduce false positives (Supplemental 2) (Zhao et al., 2019).

LAMP primer sets combination and multiplex RT-LAMP reaction

Real-time RT-LAMP assays with different combination of LAMP primer sets were performed using the SARS-CoV-2 genome RNA standard (448 copies) as the template. For duplex RT-LAMP reactions that containing two different sets of LAMP primers, our results showed that the primer combination of the set3 and set1 (S3+S1) was better than the singleplex (S3) or the primer combination of the set3 and set5 (S3+S5) in peak time and signal strength (Fig. 3A). To avoid the possibility of increasing sensitivity caused by higher concentrations of the LAMP primer sets, double the amount of set3 (S3+S3) was also included and compared, the results showed that (S3+S1) was also better than (S3+S3) (Fig. 3A). For triplex RT-LAMP reactions that containing three different sets of LAMP primers, our results showed that the primer combination of the set3, set1, and set5 (S3+S1+S5) performed better than the duplex primer combination of the set3 and set1 (S3+S1) or the singleplex (S3) (Fig. 3B). All results indicated that the triplex primer combination of the set3, set1, and set5 (S3+S1+S5) exhibited the best performance and was therefore selected as the optimal LAMP primer combination for further studies (Ji et al., 2021).

Table 2 Primer sets used for RT-LAMP assays in this study.

Primer set	Primer name	Primer sequence (5′–3′)	Target gene	Refs	
Set1	F3	TGCAACTAATAAAGCCACG	ORF1ab	Dong et al. (2021), Park et al. (2020)	
B3	CGTCTTTCTGTATGGTAGGATT	
FIP	TCTGACTTCAGTACATCAAACGAATAAATACCTGGTGTATACGTTGTC	
BIP	GACGCGCAGGGAATGGATAATTCCACTACTTCTTCAGAGACT	
LF	TGTTTCAACTGGTTTTGTGCTCCA	
LB	TCTTGCCTGCGAAGATCTAAAAC	
Set2	F3	TCACCTTATGGGTTGGGA	ORF1ab	Nawattanapaiboon et al. (2021)	
B3	CAGTTGTGGCATCTCCTG	
FIP	CGTTGTATGTTTGCGAGCAAGATTTTGAGCCATGCCTAACATGC	
BIP	GTGCTCAAGTATTGAGTGAAATGGTTTTTATGAGGTTCCACCTGGTT	
LF	ACAAGTGAGGCCATAATTCTAAG	
LB	GTGTGGCGGTTCACTATATGTT	
Set3	F3	GCCAAAAGGCTTCTACGCA	N	Dong et al. (2021), Park et al. (2020)	
B3	TTGCTCTCAAGCTGGTTCAA	
FIP	TCCCCTACTGCTGCCTGGAGGCAGTCAAGCCTCTTCTCG	
BIP	TCTCCTGCTAGAATGGCTGGCATCTGTCAAGCAGCAGCAAAG	
LF	TGTTGCGACTACGTGATGAGGA	
LB	ATGGCGGTGATGCTGCTCT	
Set4	F3	CCAGAATGGAGAACGCAGTG	N	Dong et al. (2021), Jamwal et al. (2021), Jiang et al. (2020)	
B3	CCGTCACCACCACGAATT	
FIP	AGCGGTGAACCAAGACGCAGGGCGCGATCAAAACAACG	
BIP	AATTCCCTCGAGGACAAGGCGAGCTCTTCGGTAGTAGCCAA	
LF	TTATTGGGTAAACCTTGGGGC	
LB	TTCCAATTAACACCAATAGCAGTCC	
Set5	F3	TGAGTACGAACTTATGTACTCAT	E	Yu et al. (2021), Zhang & Tanner (2021)	
B3	TTCAGATTTTTAACACGAGAGT	
FIP	ACCACGAAAGCAAGAAAAAGAAGTTCGTTTCGGAAGAGACAG	
BIP	TTGCTAGTTACACTAGCCATCCTTAGGTTTTACAAGACTCACGT	
LF	CGCTATTAACTATTAACG	
LB	GCGCTTCGATTGTGTGCGT	
Set6	F3	CCGACGACGACTACTAGC	E	Broughton et al. (2020), Dong et al. (2021)	
B3	AGAGTAAACGTAAAAAGAAGGTT	
FIP	CTAGCCATCCTTACTGCGCTACTCACGTTAACAATATTGCA	
BIP	ACCTGTCTCTTCCGAAACGAATTTGTAAGCACAAGCTGATG	
LF	TCGATTGTGTGCGTACTGC	
LB	TGAGTACATAAGTTCGTAC	
Notes.

F3/B3: outer primers; FIP/BIP: forward and backward internal primers; LF/LB: forward and backward loop primers.

Figure 2 Comparison of the performance of SARS-CoV-2 real-time RT-LAMP assays with different LAMP primer sets.

(A) Six LAMP primer sets targeting different regions of SARS-CoV-2 genome; (B) LAMP primer sets targeting the same regions (ORF1ab, N and E genes) of SARS-CoV-2 genome; NTC means DEPC-H2O.

Figure 3 Comparison of the performance of SARS-CoV-2 real-time RT-LAMP assays with different combinations of LAMP primer sets.

(A) Duplex combinations of LAMP primer sets; (B) Triplex combinations of LAMP primer sets; NTC means DEPC-H2O.

Visual detection of the SARS-CoV-2 RT-LAMP reaction

To develop easy-to-use colorimetric RT-LAMP assays, two different indicators, cresol red (a pH-sensitive indicator) and HNB (a metal-ion indicator), were included and assessed in the RT-LAMP reactions. All RT-LAMP reactions with an indicator were performed at 65 °C for 35 min. A positive reaction with cresol red yielded color change from pink to yellow, while a positive reaction with HNB exhibited color change from violet to blue (Figs. 4B and 5B). These results indicate that both the HNB and cresol red indicators are suitable for colorimetric detection in the RT-LAMP reactions.

Figure 4 Sensitivity tests of SARS-CoV-2 with the developed triple-target RT-LAMP assay.

(A) Results of the triplex real-time RT-LAMP assay, each dilution of samples were performed in triplicate; (B) Results of the triplex colorimetric RT-LAMP assay; NTC means DEPC-H2O.

Figure 5 Specificity tests of SARS-CoV-2 with the developed triplex RT-LAMP assay.

(A) Results of the triplex real-time RT-LAMP assay; (B) Results of the triplex colorimetric RT-LAMP assay; Samples and tubes: 1: hPIV-1; 2: hPIV- 2; 3: hPIV-3; 4: H1N1-pdm09; 5: H1N1; 6: H3N2; 7: infB; 8: RSV-A; 9: RSV-B; 10: hCoV-HKU1; 11: hCoV-NL63; 12: hCoV-OC43; 13: hCoV-229E; NC: DEPC-H2O; PC: SARS-CoV-2 genome RNA standard (448 copies).

Sensitivity of the SARS-CoV-2 triple-target RT-LAMP assay

Sensitivity was determined by testing serial dilutions of SARS-CoV-2 genome RNA standards with the developed triplex real-time RT-LAMP and colorimetric RT-LAMP assays. In the triplex real-time RT-LAMP assay, all of the positive amplification curves (S-shaped) appeared within 35 min when using templates ranging from 448 to 11 copies per reaction (Fig. 4A). At the same time, positive reactions with color change also occurred with templates ranging from 448 to 11 copies per reaction in the triplex colorimetric RT-LAMP assays (Fig. 4B). All results suggested similar sensitivity of the triplex real-time RT-LAMP and colorimetric RT-LAMP assays, which were able to detect down to 11 of copies SARS-CoV-2 RNA per 25 µl reaction, with higher sensitivity than the previously reported SARS-CoV-2 RT-LAMP assays (Dong et al., 2021; Luo et al., 2022).

Specificity of the SARS-CoV-2 triple-target RT-LAMP assay

First, the specificity of these six sets of LAMP primers had been well studied in the previous studies reported by the different laboratories (Broughton et al., 2020; Dong et al., 2021; Jamwal et al., 2021; Jiang et al., 2020; Nawattanapaiboon et al., 2021; Park et al., 2020; Yu et al., 2021; Zhang & Tanner, 2021). Second, the sequence of the LAMP primers was compared to aligned sequences of some other coronaviruses (including MERS-CoV, SARS-CoV, HCoV-HKU1, HCoV-OC43, HCoV-NL63, and HCoV-229E), all of which had some nucleotides mismatching with our LAMP primers, supporting the specificity of the developed RT-LAMP assay. Third, the specificity was evaluated with isolated RNA of some common respiratory viruses. Our results indicated that positive results were only observed in reactions with the presence of SARS-CoV-2 RNA as the template, and no cross reactions were detected by the triplex real-time RT-LAMP assay (Fig. 5A) and the triplex colorimetric RT-LAMP assay (Fig. 5B) with RNA isolated from clinical positive samples with other common respiratory viruses. In the triplex colorimetric RT-LAMP assay containing cresol red, although the tube 11(HCoV-NL63), tube 12(HCoV-OC43), and tube 13(HCoV-229E) shown different colors from the tubes with other negative samples, the color of these three tubes are very different from positive tube (Fig. 5B). The above results indicated that the developed triplex real-time RT-LAMP and colorimetric RT-LAMP assays are highly specific for SARS-CoV-2 detection.

Discussion

The global COVID-19 pandemic has lasted for more than two years and is likely to coexist with us for a long time (WHO, 2021). Currently, there are no effective therapies for COVID-19 or anti-viral drugs against SARS-CoV-2, so early detection of the virus is essential to contain the spread of SARS-CoV-2 (Li & De Clercq, 2020). Most available SARS-CoV-2 diagnostic tests or kits are based on the real-time RT-PCR platform, but these assays require a certified and highly specialized laboratory with well-trained personnel and sophisticated experimental equipment, which usually take more than 4 h to obtain results. These issues hamper the use of PCR-based methods. Developing more rapid and convenient assays for SARS-CoV-2 detection is of vital importance.

SARS-CoV-2 constantly changes through genetic mutations, with novel variants of concern (VOC) occurring over time, such as Alpha, Beta, Gamma, Delta and Omicron, which are more transmissible, more pathogenic, or have better capability for immune escape (Hacisuleyman et al., 2021; Hoffmann et al., 2021; Wang et al., 2021). All of these VOCs and some other genomic mutations make it difficult to detect SARS-CoV-2 with only one single target (Ji et al., 2021; Mohon et al., 2020). Therefore, LAMP primer sets targeting different SARS-CoV-2 regions may provide more accurate diagnosis results. Furthermore, all of these VOCs usually mutated in the Spike(S) gene, and in our studies, we chose ORF1ab, E and N genes of SARS-CoV-2, which were the most conserved regions within the SARS-CoV-2 genome. Moreover, compared to performing multiple singleplex RT-LAMP reactions, multiplex RT-LAMP assays reduce the cost and time for two or more targets being simultaneously amplified in one reaction (Ji et al., 2021; Kim et al., 2019; Mohon et al., 2020).

In our studies, we successfully developed a triple-target colorimetric RT-LAMP assay for SARS-CoV-2 detection within 45 min. It’s not easy for us to develop the triplex RT-LAMP assays since more primers containing in the same reaction. However, all of the LAMP primer sets were well studied by other laboratories; also, SARS-CoV-2 with large genome size (30 kb), so it’s possible for us to choose different target regions (ORF1ab, E and N genes) within SARS-CoV-2 genome.

Compared with other reported studies, our developed triple-target colorimetric RT-LAMP assay differs in several ways. First, our assays represent the first triple-target RT-LAMP assay that can detect three different genes (ORF1ab, E, and N) of SARS-CoV-2 in one reaction. Second, our assays are more sensitive than the most reported RT-LAMP assays, detecting down to 11 copies per 25 µl reaction (Dong et al., 2021; Luo et al., 2022). Third, our assay can be visualized using two different indicators (HNB or cresol red), which can be easily observed by the naked eye (Dong et al., 2021; Luo et al., 2022; Rabe & Cepko, 2020).

In most of the reported studies, the target gene segments of SARS-CoV-2 were first constructed from either in vitro synthesized DNA or PCR amplified products. RNA samples were generated by an in vitro transcription reaction, followed by the determination of their concentration and copies number. Finally, serial dilutions of the RNA standard samples were prepared and used as the templates for further studies. These processes are complicated, time-consuming, and costly (Dong et al., 2021; Luo et al., 2022). Furthermore, RT-LAMP assays targeted different SARS-CoV-2 gene regions, making it difficult to study the sensitivity of each assay using one synthesized RNA template (Dong et al., 2021). Moreover, different laboratories prepared different SARS-CoV-2 RNA standard samples which were used for sensitivity studies, so the sensitivity is unreliable and incomparable (Dong et al., 2021). In this study, we used a whole genome SARS-CoV-2 RNA standard and quantified the copy number of the SARS-CoV-2 RNA with digital droplet PCR, allowing us to easily compare and accurately assess the sensitivity of SARS-CoV-2 RT-LAMP assays developed by the different laboratories.

The developed real-time RT-LAMP assays and colorimetric RT-LAMP assays eliminate the possibility of cross contamination by avoiding opening the reaction tube, which is also one of the biggest concerns for LAMP applications. The results of colorimetric RT-LAMP assays can be easily observed by the naked eyes. Also, as the real-time RT-LAMP assays monitored fluorescent signals generating by SYBR Green, we could easily optimize the LAMP assays with amplification curves. Moreover, melting curve analyses always followed real-time RT-LAMP assay, which could be conveniently used for LAMP product analyses. All of the RT-LAMP assays presented in our studies were performed using either real-time RT-LAMP assays, or colorimetric RT-LAMP assays or both.

Sensitivity and specificity are two critical parameters for a diagnosis tool. The developed triplex real-time and colorimetric RT-LAMP assays, which can detect down to 11 copies of SARS-CoV-2 RNA per reaction, were more sensitive than the previously reported RT-LAMP assays (Dong et al., 2021; Luo et al., 2022). Furthermore, previous studies reporting Ct>35 could be used as cut-off for SARS-CoV-2 infectivity. Therefore, the developed RT-LAMP assay would be useful for the detection of highly infectious cases of COVID-19 in the field (Kampf, Lemmen & Suchomel, 2021; Platten et al., 2021). The specificity of the developed triplex real-time and colorimetric RT-LAMP assays was evaluated with RNA isolated from positive clinical samples with other common respiratory viruses, and our results indicated that positive results were only observed in reactions with SARS-CoV-2 RNA as the template, these results indicated that the developed triplex RT-LAMP assays are highly specific for SARS-CoV-2 detection.

Our developed triplex RT-LAMP assays may have potential limitations. For example, multiplex RT-LAMP assays are not easily validated and optimized. Since 18 primers are used in the triplex RT-LAMP reaction, it is essential that the LAMP primers included in the multiplex LAMP assay do not interfere with each other. Moreover, all experiments presented here used SARS-CoV-2 genome RNA standard as the templates, and our developed real-time and colorimetric RT-LAMP assays were not validated on SARS-CoV-2 positive clinical samples. Furthermore, the RT-LAMP assay is not a quantitative test, viral loads in samples were difficult to quantify using the RT-LAMP assays (Dao Thi et al., 2020).

Conclusions

In conclusion, a rapid and convenient triple-target colorimetric RT-LAMP assay was developed for SARS-CoV-2 detection. The assay has high specificity and sensitivity and may provide a useful tool for SARS-CoV-2 screening in resource-limited regions.

Supplemental Information

Supplemental Information 1 Raw data

LC96P files can be opened and analyzed with LightCycler® 96 SW 1.1 software (roche.com).

Click here for additional data file.

Supplemental Information 2 Blasting results of the LAMP primer sets

Click here for additional data file.

Supplemental Information 3 Reaction times optimization of the RT-LAMP reaction

Click here for additional data file.

We are grateful to Zhu Zhu and Isadora Zhang for critical comments and technical assistance.

Additional Information and Declarations

Competing Interests

Author Contributions

Data Availability

The authors declare there are no competing interests.

Zhu Yang conceived and designed the experiments, analyzed the data, prepared figures and/or tables, authored or reviewed drafts of the article, and approved the final draft.

Nicole Y. Liu performed the experiments, analyzed the data, authored or reviewed drafts of the article, and approved the final draft.

Zhiwei Zhu performed the experiments, analyzed the data, prepared figures and/or tables, and approved the final draft.

Minmin Xiao analyzed the data, authored or reviewed drafts of the article, and approved the final draft.

Shuzhi Zhong analyzed the data, authored or reviewed drafts of the article, and approved the final draft.

Qiqi Xue performed the experiments, prepared figures and/or tables, and approved the final draft.

Lina Nie performed the experiments, prepared figures and/or tables, and approved the final draft.

Jinhong Zhao conceived and designed the experiments, analyzed the data, prepared figures and/or tables, authored or reviewed drafts of the article, and approved the final draft.

The following information was supplied regarding data availability:

The raw data is available in the Supplementary Files.

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
