# Peer review of "Rapid and convenient detection of SARS-CoV-2 using a colorimetric triple-target reverse transcription loop-mediated isothermal amplification method"

_PeerJ, doi:10.7717/peerj.14121_

## Round 0.1 · original submission · Major Revisions

As you will see from the reviews enclosed, all three experts conclude your paper has merit and is potentially worthy of publication. However, a key question raised by them is that of sensitivity. You must address this point carefully and fully in revision.

All other points should be relatively straightforward to address with editorial work and some additional reconfiguration/data. Please explain clearly your response to each and every comment raised.

Reviewer 1 ·

Basic reporting

-

Experimental design

-

Validity of the findings

-

Additional comments

In this work, the authors developed the triple target RT-LAMP assay to simultaneously detect three different target regions of SARS-CoV-2, which exhibited the high detection sensitivity (down to 11 copies of SARS-CoV-2 RNA in 25 μL reaction). Overall, the manuscript was well written and organized. In addition, the proposed RT-LAMP assay targeting three different regions in SARS-CoV-2 genome was fully validated to provide the better detection sensitivity and selectivity compared to the previous counterpart. Therefore, I suggest the acceptance of this manuscript after accommodating the following comments.

1) The authors need to explain the reason for higher copy number of N gene compared to ORF1ab and E genes (Table 1). In addition, the copy numbers of SARS-CoV-2 were presented based on the copy number of ORF1ab gene, not N or S genes, which should be explained.
2) In line 148, it was written that “35 min was selected as the optimal reaction time in order to reduce false positives (data not shown)”. It would be more informative if the data could be provided.
3) In line 157, the set3 and set5 (S3 + S1) should be corrected into the set3 and set1 (S3 + S1).
4) In Figure 2 and 3, the authors need to provide the results in the absence of targets (NTC).
5) In Figure 2~4, the authors need to specifically indicate what each figure is.
6) In Figure 5 (pH Indicator), the samples (11-13) showed different colors from other negative samples. It would be more informative to explain this difference.
7) Recent works on the LAMP-based biosensors (e.g., Analytica Chimica Acta, 1205, 339781 (2022)) or SARS-CoV-2 detection assays (e.g., Biosensors and Bioelectronics, 208, 114221 (2022)) should be included as new references.

Reviewer 2 ·

Basic reporting

Here, Yang et al. reported a study, the main goal of which is to develop a new method for COVID-19 detection. There is an unmet need for this tool because the current RT-PCR based real-time assay needs multiple experimental steps and the usage of expensive reagents. They assessed the potential usage of the reverse transcription loop-mediated isothermal amplification method (RT-LAMP) assay. They provided reasonable data to indicate that their new assay could deliver similar sensitivity results to the conventional RT-PCR assay.

Experimental design

The experimental design is suitable for evaluating the sensitivity of their new assay. However, besides this aspect, another critical prerequisite of detection tools is sensitivity. There is a lack of sufficient information for this point of evaluation.

It would be a plus that they could include additional experimental analysis to evaluate their PR-LAMP for the current variant strains of SARS2 virus.

Validity of the findings

Overall, their results support their claim about the potential usage of RT-LAMP as an alternative detection tool.

Additional comments

None.

Reviewer 3 ·

Basic reporting

Important:

1. One of the authors’ key claims is that the triple-target RT-LAMP assay is more sensitive than single-target or double-target RT-LAMP assays. As described in the Methods, line 103, the triple-target and double-target assays include, not merely more types of primers, but an overall higher concentration of primers. Therefore, it is possible that a singleplex assay with an equivalently high primer concentration would have the same increased sensitivity as the triplex assay. This possibility may be addressed with the sample “S3+S3” in the top panel of Figure 3, but that result is never discussed in the text.

Moderately important:

2. The legend of Figure 2 should explain the relationship between the top and bottom panels.

3. The identity of all the samples in Figure 4 should be explained in the legend or on each panel. It appears that there are negative controls present in some panels but not others. I also assume that the multiple lines in each panel are replicates, but this should be stated in the legend.

4. In lines 255-258, the authors state that the 18 primers included in the triple target assay may make validation and optimization difficult. However, haven’t the authors done that in this study?

5. The paragraph at lines 234-242 describes at least three important advantages of this triple-target RT-LAMP assay. Describing these advantages in the same paragraph is somewhat confusing.

6. It would be helpful to add a very short explanation of how HNB and cresol red detect DNA amplification.

7. At lines 146-149, it would be helpful to describe what reaction characteristics were optimized for.

Minor textual suggestions (basically copy edits):

8. The sentence in lines 26-29 should be broken into two or more sentences.

9. Line 31, the , should be ;

10. Line 31, “were” should be “are.”

11. Line 32, “eyes-to-see” should be deleted.

12. Line 33, “the” at the end of the line should be omitted.

13. Line 34, “resources” should be “resource.”

14. Line 49, “acids” should be “acid.”

15. Line 54, “costs” should be “cost.”

16. Line 66, “duo” should be “due.”

17. Line 69, it is unclear which reaction the clause that begins “which” is referring to.

18. Line 71, “our” should begin a new sentence.

19. Lines 120-121: This sentence is unclear.

20. Line 132, did the authors design the primers or were they from the literature? This should be made clear.

21. Line 136, it should be mentioned to which SARS-CoV-2 genome or genomes the primers matched.

22. Lines 141-143: This sentence is unclear.

23. Line 145, “combinations” should be “combination.”

24. Line 190, “presence SARS-CoV-2” should be “presence of SARS-CoV-2.”

25. Line 197, “likely coexist” should be “likely to coexist.”

26. Line 207, “or better” should be “or have better.”

27. Line 210, “targeted” should be “targeting.”

28. Line 267, “with” should be “has.”

29. Line 267, the comma should be replaced with “and.”

30. Line 268, “the” should be removed.

Experimental design

31. If point 1 above has not been addressed in the experiments, it should be addressed, or that should be explained in the text.

Validity of the findings

Important:

32. At several points in the paper, including line 68, the authors claim that the triple-target assay “showed a higher sensitivity than the singleplex or duplex RT-LAMP assay previously reported…” This claim should include a citation at line 68-69. The claim is made again at lines 178-179, where it does include a citation. However, more explanation is required. This paper reports that as few as 11 copies of the SARS-CoV-2 genome can be detected by the triple target assay. Dong et al. 2021 reports that they can detect as few as 8 copies and Luo et al. 2022 reports a limit-of-detection of 10 copies. The two previous publications used different and possibly inferior standards for determining the limit of detection, as described in lines 223-233 of this paper. Nevertheless, claims of higher sensitivity should either be removed from the paper or much better supported. (If I am misinterpreting, the authors should make their sensitivity comparisons more clear, as other readers are likely to make the same mistake I did.)

Additional comments

Overall, this manuscript is well-written and, in general, the data support the conclusions. It is likely that only textual changes are needed. The claims of higher sensitivity relative to previously published assays should be clarified. Whether higher sensitivity is due to having three targets or merely higher primer concentration should be determined or clarified.

---

## Round 0.2 · accepted · Accept

Thank you for addressing the points. I am pleased to recommend acceptance.

Reviewer 2 ·

Basic reporting

Authors have addressed my main concerns.

Experimental design

None.

Validity of the findings

None.

Additional comments

None.